# On the Choice of the Extracellular Vesicles for Therapeutic Purposes

**DOI:** 10.3390/ijms20020236

**Published:** 2019-01-09

**Authors:** Claudia Campanella, Celeste Caruso Bavisotto, Mariantonia Logozzi, Antonella Marino Gammazza, Davide Mizzoni, Francesco Cappello, Stefano Fais

**Affiliations:** 1Department of Biomedicine, Neuroscience and Advances Diagnosis (BIND), Section of Human Anatomy, University of Palermo, 90127 Palermo, Italy; claudiettacam@hotmail.com (C.C.); celestebavisotto@gmail.com (C.C.B.); antonella.marino@hotmail.it (A.M.G.); francapp@hotmail.com (F.C.); 2Euro-Mediterranean Institute of Science and Technology (IEMEST), 90139 Palermo, Italy; 3Institute of Biophysics, National Research Council, 90143 Palermo, Italy; 4Department of Oncology and Molecular Medicine, National Institute of Health, 00161 Rome, Italy; mariantonia.logozzi@iss.it (M.L.); davide.mizzoni@iss.it (D.M.); 5Department of Biology, Temple University, Philadelphia, PA 19122-6078, USA

**Keywords:** extracellular vesicles (EVs), exosomes, biomarkers, nanodelivery, theranostics, regenerative medicine

## Abstract

Extracellular vesicles (EVs) are lipid membrane vesicles released by all human cells and are widely recognized to be involved in many cellular processes, both in physiological and pathological conditions. They are mediators of cell-cell communication, at both paracrine and systemic levels, and therefore they are active players in cell differentiation, tissue homeostasis, and organ remodeling. Due to their ability to serve as a cargo for proteins, lipids, and nucleic acids, which often reflects the cellular source, they should be considered the future of the natural nanodelivery of bio-compounds. To date, natural nanovesicles, such as exosomes, have been shown to represent a source of disease biomarkers and have high potential benefits in regenerative medicine. Indeed, they deliver both chemical and bio-molecules in a way that within exosomes drugs are more effective that in their exosome-free form. Thus, to date, we know that exosomes are shuttle disease biomarkers and probably the most effective way to deliver therapeutic molecules within target cells. However, we do not know exactly which exosomes may be used in therapy in avoiding side effects as well. In regenerative medicine, it will be ideal to use autologous exosomes, but it seems not ideal to use plasma-derived exosomes, as they may contain potentially dangerous molecules. Here, we want to present and discuss a contradictory relatively unmet issue that is the lack of a general agreement on the choice for the source of extracellular vesicles for therapeutic use.

## 1. Introduction 

Cells are able to communicate with each other and with the surrounding environment through direct contact or the secretion of soluble factors [1,2,3,4]. The three major types of communication that cells use are active transport, passive transport, and vesicular transport [5,6]. This has generated in medicine the idea to exploit the natural system for cellular communication with therapeutic purposes. In particular, the intercellular transfer of molecular and genetic material through extracellular vesicles (EVs) has aroused considerable attention in recent years [3,7,8]. In fact, EVs represent an evolutionarily conserved mechanism to transfer biologically active molecules between cells locally and at distance, thereby regulating gene expression and cellular function in recipient cells. EVs are heterogeneous in origin, size, antigenic composition, and functional properties [7]. It is intriguing that microenvironmental conditions may change both the EVs’ composition at both protein and lipid levels, and the amount of EVs released, as far as cancer is concerned [9,10,11]. EVs are small phospholipid bilayer vesicles released by all prokaryotic and eukaryotic cells, including cancer cells, which can contain different types of RNA, proteins, mitochondrial DNA, and both single strand DNA and double strand DNA [7,12,13,14]. EVs can be classified according to their size, mode of biogenesis, functions, and composition into categories based: Exosomes (20–150 nm of diameter), apoptotic bodies (>800 nm), microparticles (0.1–1 μm), prostasomes (50–500 nm), and tolerosomes (~40 nm) [12,15]. Many recent reports have focused on the use of EVs, among these the exosomes in particular, as biomarkers for early diagnosis and as accurate therapeutic agents for various pathologic conditions, such as inflammation, cancer, and cardiovascular disease [9,16,17,18]. Considering the plethora of research on personalized therapies has involved EVs as well, in particular, those of nanosize, i.e., exosomes, through this work, we will predominantly focus the attention on exosomes, because they are currently the best characterized EVs. Effectively, for their own features, exosomes are easily accessible and capable of representing their parental cells, and these properties, for instance, may be exploited to overcome the most critical issue of regenerative medicine, such as the invasiveness and safety of therapies. Nevertheless, much remains to be made in this field of research. Indeed, the current efforts of researchers and clinicians are aimed to better characterize and to the engineering of these vesicles in order to modify their content and to use them as delivery systems for therapeutic purposes [19].

Cells produce a wide spectrum of EVs, which are believed to serve various functions depending on their origin and molecular composition. Among to the heterogeneous group of EVs, there are two main types of phospholipid vesicles, which have been classified in microvesicles (50–1000 nm) and are generated by outward budding of the plasma membrane, and exosomes (20–150 nm), generated by invagination of endosomal membranes and by the subsequent release of the multivesicular bodies (MVBs) [20,21]. Both EVs may help cells to dispose of cellular material and transfer signaling molecules, such as miRNAs, mRNAs, proteins, and lipids, to specific target cells.

Each class of EVs has specific markers, for example, tetraspanins, such as Alix, Tsg101, CD9, CD63, CD81, and CD82, are typical exosomal markers, together with heat shock proteins (Hsps) and MHC molecules as reported in the main EVs databases [22,23]. However, the increasing interest regarding the application of EVs in therapies prompted the discovery of new potential markers [24,25]. Exosomes and microvesicles have particular biological functions [26,27,28], but, due to the technical limitations in the methods of isolation and characterization, the term “extracellular vesicle” (EV) is often used generically to indicate all the classes together [21]. EVs are active players in intercellular communication mechanisms [11,27,29,30]. All cell types (including T cells, B cells, dendritic cells, platelets, epithelial cells, and cancer cells) are able to release EVs, such as exosomes, into the extracellular environment in vitro and in vivo, in both normal and pathological conditions. In fact, exosomes have been detected in virtually all biological fluids, including urine, breast milk, plasma, saliva, cerebralspinal fluid, amniotic fluid, seminal plasma, and bronchoalveolar lavage fluid [7,12,31,32,33,34,35,36,37,38,39]. It is believed that exosomes have a biogenesis mechanism that involves the endosomal pathway. After endocytosis, the early endosomes become part of the multivesicular bodies (MVBs), undergoing an unconventional inward budding and gradual manipulation of the content. Thereafter, the intraluminal vesicles (ILVs) of the MVBs, following the merging of them with the plasma membrane, are released in extracellular milieu and became exosomes [20].

The endosomal sorting complexes required for transport (ESCRT)-I play a central role in MVB formation, sorting, and secretion [40], because it has been shown to assist the sorting of cargo proteins at the endosome membranes and ALIX (apoptosis-linked gene 2-interacting protein X) is a key regulator of this function [41]. However, some evidence assumes the intervention of ESCRT-independent mechanisms of exosomes’ release, which require sphingolipid ceramide and depends on raft-based microdomains [42]; probably used for sorting of proteolipid molecules. The pathways might not be entirely separated and both pathways might work together [43,44,45].

Exosomes are easily accessible and their content reflects the characteristics of their cellular source. This, of course, allows monitoring of exosomes obtained from human body fluids while this approach is not entirely safe due to the uncompleted knowledge on the exosome composition. Of course, it is an exciting challenge for scientists to both better characterize and possibly modify the composition of these vesicles, making natural exosomes a promising delivery system for therapeutic purposes [19]. In the following paragraphs, we describe the current knowledge and possible perspectives in exosome application for disease treatment, including regenerative medicine.

## 2. EVs: Nano-Sized Carrier of Cellular Messages in Pathological Conditions

Many reports suggest that the EVs’ content is dependent on the cellular source [9,12,14,46,47,48]. Because they have an ability to bind target cells and/or exchange molecules, they can modulate the activity of other cells [9], thus exosomes, for instance, carry and deliver information that is essential for health, and participate in the activation of the immune system and in pathological events, including malignant transformation [30,49,50]. These vesicles have been identified in the plasma and in the serum of patients with different types of tumor, such as glioblastoma [51], lung cancer [52], melanoma [53,54], prostate cancer [10,55], and colon-rectal cancer [33,56,57]. Many data support the hypothesis that cancer cells can secrete more exosomes than healthy cells, and, more importantly, the lipid, nucleic acid, and protein content of exosomes are tumor-specific, and this effect is related to the tumor extracellular acidity [10,12]. This, in turn, suggests that exosomes contain stimulatory and inhibitory components that, when delivered to the recipient cells, enable crosstalk between tumor cells and their surrounding environment [58,59]. Among the molecules contained in the exosomes, special attention has been given to miRNA, which can modulate the gene expression of target cells also very far from the site of the tumor [60]. Other findings demonstrate how exosomal miRNAs can be important mediators of inflammatory responses, such as in asthma [61], diabetic nephropathy [62], and in dementia progression [63]. In addition, many reports have shown that circulating exosomes could be changed in number and composition upon cardiac injury, such as myocardial infarction, myocardial ischemia-reperfusion injury, atherosclerosis, hypertension, and sepsis cardiomyopathy [64,65,66]. However, plasmatic exosomes may represent the shuttle of potentially dangerous molecules in healthy conditions as well. As an example, exosomes may deliver both Epstein-Barr virus (EBV)-related proteins and nucleic acids [67]; but also prion proteins that are normally detectable in human exosomes, but that may contain prion-disease-related proteins as well [68].

The exosomes influence the immune system and exert physiological and pathological effects through mechanisms not completely understood. A pioneer paper was in 1996 from Raposo et al. [1], showing, through an in vitro study, how the B lymphocytes secrete antigen-presenting vesicles, capable of triggering potent T cells responses [1]. Probably, exosomes are able to stimulate naïve CD4^+^ T cells both in vitro and in vivo conditions, by the transfer of MHC-peptide complexes to denditic cells (DCs), conferring them a maturation signal necessary to naïve T cell priming [69]. Almost all cells of both innate and adaptive immunity are able to secrete exosomes that carry, for instance, MHC class I or II-peptide complexes as a mode of antigen presentation [70]. These exosomes represent a critical component of the immunoregulatory network and may help to distribute antigens, favor the induction of DCs maturation and promote secretion of pro-inflammatory cytokines, and induce a proper immune reaction against pathogens [71].

Evidence is accumulating that exosomes induce cellular-source-related functions. In fact, mature DCs release exosomes, which have the effect to promote pro-inflammatory functions [69], while many tumor cells release EVs with anti-inflammatory and immune escape effects [29,33,47,48,72]. It appears therefore conceivable that EVs and exosomes may be involved in the pathogenesis of several human diseases [72,73,74,75]. 

During tumorigenesis, tumor cells release exosomes that are not only a cargo to deliver molecules considered generally as biomarkers, but also factors that have the capability to alter phenotypic and functional attributes of recipient cells, favoring metastasis, tumor progression, and immunosuppression [76,77,78]. Tumor cells, in fact, secrete immunologically active exosomes with pro-inflammatory and/or immune suppressive factors. Then, tumor exosomes are involved in the tumor immune escape through different mechanisms, including the direct killing of anti-tumor lymphocytes through either a death-receptors-ligand interaction [79,80] or inhibiting the Natural Killer (NK) cells function [81]. A study performed with NK-cell derived exosomes has shown that NK-cell release exosomes that express both typical NK cell markers and killing molecules, and a comparable pattern has been shown in exosomes purified from the plasma of healthy individuals [82]. Notably, NK-cell derived exosomes induce the killing of target cells, and this was demonstrated with both exosomes purified from supernatants of NK-cell culture and those purified from human plasma [82]. The ensemble of these results triggered further studies, showing that exosomes secreted by tumour cells and engineered to express specific molecules could improve antitumor immunity [29]. 

## 3. Therapeutic Potential of EVs and Exosomes: From Cancer to Regenerative Medicine

In the last decade, many efforts have been made in order to set up stable and bio-compatible carriers for the delivery of chemical drugs and small molecules to be used in the treatment of human diseases. However, the Food and Drug Administration (FDA) have approved few of these [83]. The reason for this slowing down in the search for effective vectors is due to the difficulties related to immunogenity and side effects. 

The use of exosomes for therapy of human disease is becoming a central issue in NanoMedicine for their ability to deliver biologically active material to target cells [84]. For their natural ability to deliver biologically active material to target cells, exosomes may be defined as a natural nano bullet with high therapeutic potential [3]. However, the effective therapeutic advantage and the safeness of their use is still controversial. Most scientific reports produced in the last years were aimed at investigating the role of exosomes in favouring some pathological conditions or tissue degeneration. Moreover, there is a vacuum in the translation of the pre-clinical studies into clinical use, as is unfortunately occurring all too frequently in the translational research in medicine. This is the case of exosome research as well and presently we have no clear data on a number of important issues, including the potential of the natural exosome content as a new therapeutic tool, and which exosome source is more suitable to be exploited as a delivery for therapeutic molecules.

The most investigated as an artificial cargo for drugs have been the liposomes [85]. The liposomes are spherical lipid bilayers synthesized easily in the laboratory, whose features are dependent on the lipids of which are constituted [83,84,85]. However, there are a number of concerns on both their real efficacy in delivering the drugs to the disease site, and, more importantly, on the side effects due to the high catabolic index [86,87,88]. It is therefore mandatory to investigate the whole potential of exosomes as a natural delivery for both chemical and biological therapeutic molecules.

There are different targeting strategies to enhance the therapeutic potential of exosomes and of extracellular vesicles. The strategies currently used are focused on two different approaches that focus, respectively, on cellular modification via engineering techniques or are centred on direct EV alteration (Figure 1). Cellular modification consists of genetic modification, metabolic labelling, and exogenous delivery, which have been proven to change the pattern of surface expression molecules and the cargo of EV [89]. In particular, it has been proven that modification of exosomal tetraspanin complexes (CD63, CD9, CD81) strongly influences target cell selection both in vitro and in vivo [90], improving targeting of exosome to tissues and the cell type of interest. This approach may lead to the creation of exosome containing modified membrane protein with signalling properties using a lentiviral vector, a reporter system useful to generate a stable integration of the target protein within the exosomal membrane to be used for in vivo and cell uptake studies [91] (Table 1). Moreover, this system has been widely used to enhance exosome therapeutic efficacy. Modified exosomes carrying modified transmembrane proteins are of great interest for the treatment of cancer, diabetes mellitus, cardiovascular disease, and neurological disorders [92]. Together with tetraspanin, other modified transmembrane proteins are the platelet-derived growth factor receptors [93], the lysosome-associated membrane glycoprotein 2b (Lamp-2b) [94], the lacthadein, and the glycosylphosphatidylinositol (GPI) [95]. For example, a display of GPI-anchored anti-epidermal growth factor (EGFR) nanobodies on extracellular vesicles promotes tumour cell targeting [96] and targeted exosomes to neurons and glia expressing Lamp-2b showed the capacity to cross the blood-brain barrier, delivering loaded siRNA for the knockdown of BACE 1 [94]. Exosomes derived from dendritic cells and bearing Lamp-2b fused to αv integrin-specific iRGD peptide (CRGDKGPDC) has been used as an efficient drug delivery system to breast cancer cells in culture and in vivo [96] (Table 1). Furthermore, using the indirect modification of dendritic cell-derived exosomes content, it has been proven that these EVs were able to induce and enhance antigen-specific T cell responses in vivo, compared to the immunotherapy [97]. Numerous studies have focused on the cytotoxicity of the modified exosomes in different types of cancer. Similarly, systematically injected exosome carrying specific EGFR peptides fused to platelet-derived growth factor receptors (PDGF-R) delivered let-7a miRNA in EGFR-expressing xenograft breast cancer tissue in RAG2^(−/−)^ mice with a therapeutic response [93] (Table 1). 

The alternative modification strategy of the EVs that should be mentioned is direct modification. This approach can be realized whether by passive loading techniques, which exploit spontaneous membrane interactions; or through physical methods that temporarily destroy the integrity of the membranes to allow cargo loading (Figure 1). Regarding passive loading, an example has been shown by Sun et al., which demonstrated the anti-neoplastic properties of curcumin loaded in the exosomes, exploiting the hydrophobic interactions between the membrane and molecular cargo [98]. Later, active modifying techniques of EVs have been applied to demonstrate that the modification has made the EVs more bioactive and bioavailable when administered in vivo [99,100] (Table 1).

The potential application of exosomes and EV in general as therapeutic agents has led to the development of new and advantageous tools for the intracellular delivery of target proteins. Recently, EXPLORs (exosomes for protein loading via optically reversible protein-protein interactions) have been described by integrating a reversible protein-protein interaction module controlled by blue light with the endogenous process of exosome biogenesis, and are able to successfully load cargo proteins into newly generated exosomes. The module is controlled by blue light and the results indicated the potential of EXPLORs as a mechanism for the efficient intracellular transfer of protein-based therapeutics into recipient cells and tissue [101].

Exosomes can efficiently deliver mRNAs, miRNAs, and siRNAs with a wide range of applications in genetic therapies and drug discovery [92,102,103] (Figure 1). In particular, miRNAs are commonly carried by exosomes, representing in the last few years the new frontiers of the therapeutic approach to different human diseases. miRNAs are easily loaded into exosomes via miRNAs’ expression of backbones or transfection of precursor or miRNAs’ mimics [92]. For example, exosomes derived from miR-146a—overexpressing dendritic cells have been shown to suppress the effects of myasthenia gravis [104] and exosomes derived from miR-140-5p-overexpressing human synovial mesenchymal stem cells enhance cartilage tissue regeneration and prevent osteoarthritis of the knee in a rat model [105]. 

In some in vivo models, the applicability of exosomes derived from stem cells (SCs) for therapeutic purposes have been shown. The SC-derived exosomes have recently demonstrated the potential to treat many diseases and disorders, such as cardiovascular ischemia [106,107,108], and kidney and liver diseases [109,110], and enhanced wound healing as well as adipose tissue regeneration [111,112]. It has been demonstrated that exosomes from cardiosphere-derived cells have limited injury and improved function in myocardial infarction [113]. Furthermore, evidence has been provided that exosomes purified from pericardial fluid of patients undergoing acute myocardial infarction are related to an improvement in myocardial performance through a framework, including EMT-mediated epicardial activation, arteriogenesis, and reduced cardiomyocyte apoptosis [114]. NK-cell derived exosomes kill target cells using either Fas-Fas-ligand interaction or perforin activity [115].

As described above, the exosomal surface and content might be modified, giving the possibility of adding molecules, drugs, or compounds, thus enhancing their therapeutic potential.

Regenerative medicine has the purposes to functionally restore tissue from damage, malfunctioning, or to form it when it is missing. There are three main approaches in regenerative medicine: Cell-based therapies, tissue engineering, and material based approaches. In cell-based therapies, cells are administered to restore a tissue either directly or through paracrine functions; tissue engineering consists of the combined use of cells and a biodegradable scaffold to produce a tissue; and material based approaches are centred on the use of biodegradable materials, often functionalized with cellular functions. [116]. Recently, research insights have suggested that the local healing process plays an important role in regenerated tissue rather than the structural contribution of stem cells per se [117,118]. For instance, it has been demonstrated that direct injection of hematopoietic stem cells (HSCs) into the mouse heart does not result in de novo cardiomyogenic events or tissue regeneration. Despite the promising therapeutic applications of SCs, heterogeneous efficacy data have been reported, probably because this approach still needs to be clearly defined and standardized [119]. Given that, researchers are increasingly focused on the paracrine hypothesis, studying the stimulating factors released by stem and progenitor cells, such as growth factors, cytokines, and EVs, particularly exosomes. As stated above, EVs are involved in angiogenesis, the regulation of immune responses, and extracellular matrix remodeling, affecting cell phenotype, recruitment, proliferation, and differentiation [1,120,121]. All these characteristics are considered of great interest for tissue engineering and in restoring function in damaged tissues.

The most striking evidence supporting the use of exosomes in cell-free therapy and tissue engineering come from the results obtained from a series of studies on mesenchymal stem cell (MSCs) transplantation for tissue regeneration [122,123].

However, other reports have shown the ability of MSCs to induce cellular modifications through the exosomes they produce, supporting the use of exosomes in regenerative medicine [122,124,125,126]. In addition, exosomes released from monocyte, leukocyte, granulocyte, and lymphocyte are involved in the recruitment of inflammatory cells, angiogenesis, and coagulation, which trigger tissue repair and regeneration [127,128]. The clinical applications include different therapeutic protocols from neuronal regeneration to myocardial, liver, kidney, muscle, skeletal, and chondral regeneration as well as the regeneration of other tissues and organs (reviewed in 115) (Figure 1). In 2013, Patel and collaborators isolated a novel kind of stem cell from the sub-epithelial layer of the umbilical cord. These cells were phenotypically MSCs (i.e., they expressed CD9, SSEA4, CD44, CD90, CD166, CD73, and CD146) [129] and released large exosomes amounts [129]. In particular, CD146 expression influences periapical cyst MSCs’ properties [24], even if there is no evidence regarding a direct effect mediated by CD146+ EVs. Interestingly, CD146+ EVs are involved in an increased risk of acute graft-vs-host disease after allografting [25].

A potent challenge is the nervous tissue repair and regeneration. For example, miR-133b contained in exosomes extracted from multipotent mesenchymal stromal cells have the properties to boost neurite outgrowth [130]. Moreover, systemically administered exosomes generated from human bone marrow mesenchymal stem cells (hBMSCs) attenuated neuroinflammation and enhanced angiogenesis and neurogenesis in a rat model of traumatic brain injury [131]. In a similar manner, exosomes derived from human adipose-derived stem cells (hASCs) showed pro-regenerative effects on neuronal cells post injury [132] and exosomes isolated from BMSCs significantly boost the survival of retinal ganglion cells via argonaute-2, promoting the regeneration of their axons [133]. Furthermore, the pro-regenerative properties of exosomes were observed also in spinal cord injuries. After nervous damage, exosomes derived from Schwann cells could be internalized by axons and drove axonal regeneration, inhibiting the activity of RhoA responsible for growth cone collapse [134]. Finally, it was demonstrated that the exosomal retinoic acid receptor β (RARβ) taken up by astrocytes could reduce their proliferation, preventing scar formation around regenerating axons [135].

Regarding myocardial regeneration, exosomes isolated from cardiosphere-derived cells could promote the proliferation of cardiomyocytes when injected into mice in a model of ischemic injury [136]. Moreover, preconditioning with MSC exosomes could boost the proliferation, migration, and angio-tube formation of cardiac stem cells [137]. Khan and colleagues reported that stem cell-derived exosomes delivered by the intra-myocardial administration in mice at the time of myocardial infarction induced neovascularization and cardiomyocyte survival, enhancing c-kit positive cardiac progenitor cells’ survival and proliferation [138]. Other research groups demonstrated that exosomes derived from human umbilical cord mesenchymal stem cells (hUCMSCs) might protect myocardial cells from apoptosis by modulating the expression of members of the Bcl-2 family, thus promoting angiogenesis [139]. 

Interestingly, Agarwal and collaborators demonstrated for the first time that donor age and hypoxia level could influence the therapeutic efficacy of exosomes derived from human cardiac progenitor cells [140]. Recently, it has been found that the pericardial fluid also contained exosomes enriched with miRNAs and that they might improve the survival, proliferation, and networking of endothelial cells in vitro. Most importantly, the exosomes in the pericardial fluid might boost flow recovery and angiogenesis in a mouse model of ischemic injury [141]. 

Exosomes exert a pivotal role also in hepatic and kidney regeneration. Exosomes are used as specific biomarkers for hepatocyte damage and inflammation in liver diseases [142]. In acute liver injury, exosomes derived from hepatocytes could promote the proliferation of hepatocytes in culture and liver regeneration in vivo [110,143]. Moreover, administration of hUCMSC-derived exosomes could effectively rescue mice from liver failure in a carbon tetrachloride (CCl_4_)-induced liver injury mouse model [144]. Regarding kidney regeneration, it has been demonstrated that human umbilical cord blood-derived endothelial colony forming cells (ECFCs) and derived exosomes intravenous administration might attenuate renal injuries in mice with ischemic injury [145]. In addition, Wang et al. found that exosomes derived from engineered MSCs overexpressing miRNA-let7c could attenuate kidney injury, achieving antifibrotic functions [146]. 

Other fields of interest for regenerative medicine are skeletal, chondral, and muscle regeneration. In particular, bone regeneration using MSCs and tissue engineering approaches is one of the most widely researched fields. Exosomes isolated from MSC-conditioned medium could accelerate femur fracture healing [147]. Moreover, Zhang et al. investigated the pro-osteogenic potential of human-induced pluripotent stem cell (hiPSC)-derived MSC-exosomes by activating the PI3K/Akt signalling pathway [148]. Recently, the therapeutic effects of exosomes derived from human embryonic mesenchymal stem cells have also been demonstrated on cartilage repair [122]. Moreover, exosomes secreted by human synovial membrane MSCs and induced pluripotent stem cell-derived MSCs exert a regenerative potential on osteoarthritis provided new perspectives for cell-free therapies for cartilage injury [149]. Regarding skeletal muscle, exosomes derived from MSCs could promote myogenesis and angiogenesis in vitro [150]. Moreover, exosomes derived from human skeletal myoblasts could induce a myogenesis response during myotube differentiation [151] and could accelerate skeletal muscle regeneration by reducing collagen deposition and increasing the number of regenerated myofibers in injured muscles [152].

A number of research groups are involved in cutaneous regeneration. For instance, angiogenesis is of crucial importance in various physiological processes, including tissue regeneration and cutaneous wound healing. Exosomes released by human adipose derived MSCs can significantly induce endothelial cell angiogenesis in vitro and in vivo, promoting the release of miR125a [153]. One of the most common causes of cutaneous damage is burn injury. Further investigation indicated that exosomes released from hUCMSC could successfully reverse the burn-induced inflammatory reaction [154]. Interestingly, exosomes derived from human amniotic epithelial stem cells, when subcutaneously injected around the wound site, promote the migration and proliferation of fibroblasts, accelerating healing of full-thickness skin defects in a dose-dependent manner [155]. In a similar manner, exosomes obtained from platelet-rich plasma and exosomes derived from corneal epithelial cells could promote angiogenesis and accelerate healing in a number of experimental models both in vitro and in vivo [156,157].

The potential of the use of exosomes in regenerative medicine is not limited to the tissue described above, with studies including pancreas and dental pulp as well [152,158]. Thus, there is an exciting background supporting the use of nanovesicles in regenerative medicine, but sadly, there is little clinical evidence. One central topic, which is the key point of this work, is the choice of the source of exosome to be exploited in the generation of new tissues for therapy.

A key point in the choice of natural nanovesicles as a shuttle for therapeutic molecules is to avoid the use of exosomes with potential toxic effects. This may be the case of plasmatic exosomes that in patients may deliver molecules administered in the current treatment schedules. Cancer patients’ exosomes may, for instance, deliver chemotherapeutics as it has been shown both in vitro and in vivo [155]. However, together with mesenchymal stem cells, monocyte-derived macrophages have also been shown to represent a safe and valuable source for therapeutic exosomes [156] 

The debate is indeed entirely open, most of all because the clinical data represent a negligible part of what has been published in exosome and EVs research. Figure 1 summarizes what we can conceivably propose on the basis of the published research. 

## 4. Conclusions and Remaining Challenges

Based on their key function of being natural mediators in cell-cell communication, EVs are promising candidates in the treatment of numerous pathologies and in tissue regeneration. As described above, exosomes may have a regenerative potential themselves, but thanks to their bioavailability and low immunogenicity, they are optimal candidates for use as carriers of drugs and therapeutic molecules. There is evidence demonstrating that heterologous exosomes released by mesenchymal stem cells may be considered as a reliable and safe source for therapeutic exosomes. However, the use of an autologous approach cannot be excluded. In this case, exosomes should be free of potentially dangerous molecules, but patients’ plasmatic exosomes, for example, are dangerous by definition. In fact, plasmatic exosomes deliver molecules that may be considered a sort of waste of the diseased tissues, but they may well deliver drugs with high toxic potential [159]. We have recently shown that exosomes released by human primary macrophages efficiently deliver drugs that may, in turn, be used as both tracers and therapeutic molecules [160], with a promising future in theranostics [161]. These studies support the use of autologous exosomes obtained by peripheral-blood derived primary monocytes. Of course, we do not mean that they are entirely safe, but they are safer than plasmatic exosomes.

Thus, there are no doubts on the efficacy of new therapies based on the use of exosomes [162,163]. Clinical investigation is, however, mandatory to establish the real therapeutic potential of EVs, but also their safety, which remains a point still under debate. 

The idea to modify exosomes to render them safer and more effective is of course an amazing endpoint (Figure 1), but before getting to this point, we should try to establish which are the safest sources of exosomes for therapeutic use. 

For example, the use of the platelets-derived exosomes are a very interesting candidate in regenerative medicine because of their low immunogenicity [156], but this does not exclude the possibility that graft versus host disease may arise. The use of exosomes isolated from the same patient, which undergoes therapies and their manipulation, would avoid all the problems of the immunogenicity, but it would greatly extend the time for therapy (Figure 1).

Another issue that needs to be addressed is the mechanism/s through which exosomes may exert their therapeutic effects.

It appears that a key mechanism used by EVs to interact with target cells is the membrane-to-membrane fusion and the delivery of the vesicles’ content within the target cells. This of course may per se trigger an effect. For instance, it has been shown that exosomes derived from different cell types may have preferential targeting towards certain cell types based on their membrane composition, thus imparting a differential effect on our body [164]. A possible mechanism that has been investigated in a tumor mice model consists of a modified gene expression profile of receiving cells, such as genes involved in down-regulated functions, such as cell death, growth, and proliferation [165,166]. One of the well-known effects is the contribution to maintain the immune homeostasis, by modulating the inflammatory expression of immune cells, both in in vitro and in vivo models [167,168]. Nevertheless, the mechanism on how the EVs function and influence their target cells remains to be clarified. It is debated whether EVs are able to restore tissue integrity and function through direct cell activity modulation or by an indirect effect on the immune system. Most likely, the complete characterization of their content may be a promising challenge to elucidate the full potential of natural nanovesicles.

To accomplish this, the optimum could be achieved by creating an “exosome-factory”, similar to cell factories for cell therapy, in which exosomes can be obtained from the same cells of the patient who must undergo treatment (i.e., MSCs or iDCs) or from other cell types, which are engineered to internalize within the exosomes’ specific therapeutic molecules (Figure 1). This may be a way to reduce costs and the delivery of therapeutic molecules easier, hopefully reducing the well-known side effects of current treatments of chronic diseases.

Given the above, whatever the modification made on the exosomes or the choice of the cell-source, the problem of their safety remains. In fact, the challenge of research in the next few years probably will be to predict the exosomes’ behavior once injected. Little wonder all this can be achieved only after a careful and intense investigation on the characterization of the content of exosomes to be approved for clinical use.

## Figures and Tables

**Figure 1 ijms-20-00236-f001:**
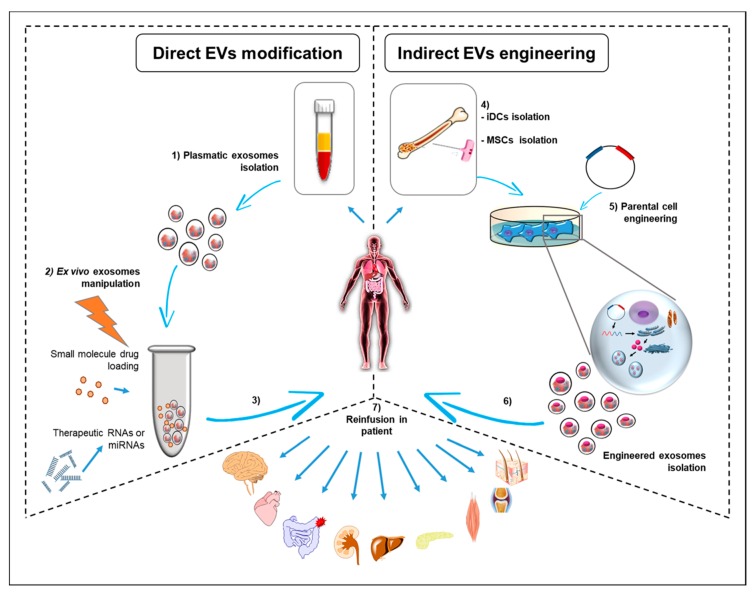
Exosomes as nanomedicine tools. Exosomes can be isolated from the patient, starting from all body fluids. The less invasive access is represented by a simple blood sample (liquid biopsy), by which it is possible to isolate circulating exosome (1), potentially released by all body cells. These “self” exosomes obtained can be loaded with drugs (2), such as peptides or small molecules, otherwise with therapeutic RNAs or miRNAs and then reinfused in the same patient for the therapeutic purpose (3). Another approach consists in the possibility to isolate and manipulate, for instance, immature dendritic cells (iDCs) or mesenchymal stem cells (MSCs) (4), with the aim of producing exosomes bearing therapeutic molecules (5). The engineered exosomes may be administered to the patient (6) as a treatment of diseases affecting various organs, or to exploit the recognized regenerative capacities (7).

**Table 1 ijms-20-00236-t001:** Experimental studies on EVs modification: Bioactivity and therapeutic implications.

EVs Cell Source	EVs Cargo	EVs Engineering Strategy	Therapeutic Effects	Ref.
Rat pancreatic adenocarcinoma cell line Rat endothelial cell line Rat lung fibro- blast cell line Rat lymph node stroma cell line Human embryonic kidney cell line	Tetraspanin chimeric-complexes	Indirect approach	Improving of targeting selection	[90,91]
Human embryonic kidney cell line	Transmembrane domain of platelet-derived growth factor receptor	Indirect approach	Targeting to xenograft breast cancer cells in RAG2^−/−^ mice	[93]
Self-derived dendritic cells	Chimeric peptide (Lamp2b-RVG; MSP; FLAG epitope) Chimeric peptide (Lamp2b-αv iRGD peptide)	Indirect approach Indirect and direct approach	Targeting to neuronal tissue and muscle tissue Enhancing of chemotherapy index	[94,96]
Mouse neuroblastoma cells	Chimeric peptide (Lamp2b-GPI-anchor peptide)	Indirect approach	Targeting of EGFR-expressing tumour cells	[95]
Dendritic cells	rAAV/AFP	Indirect approach	Enhancing antigen-specific T cell responses in vivo against cancer	[97]
Mouse lymphoma cell line Murine macrophage cell line	Curcumin	Direct approach	Protection against lipopolysaccharide (LPS)-induced septic shock in mouse	[98]
Human cervical cancer-derived) cells Human epidermoid carcinoma cells Mouse neuroblastoma cells	Oligoarginine peptides Targeting ligands conjugated to PEG	Direct approach	Improving of therapeutic effects and in vivo bioavailability	[99,100]

Abbr.: RAG2, recombination activating gene 2; RVG, rabies viral glycoprotein peptide; MSP, muscle-specific peptide; GPI, glycosylphosphatidylinositol; iRGD, integrin-specific; rAAV/AFP, recombinant adeno-associated viral vector -carrying alpha-fetoprotein gene; PEG, polyethylene glycol.

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
