# Peer review of "On the Choice of the Extracellular Vesicles for Therapeutic Purposes"

_ijms, 2019, doi:10.3390/ijms20020236_

Round 1
Reviewer 1 Report
The work of Fais et al describes the use of EVs as therapeutic approach. The Review is interesting however few comments need to be addressed.
1) A specific paragraph related to the use of EVs as emerging targets for drug delivery should be reported as EVs have been used for the delivery of biologic, drugs, small molecules
2) Since this Review is about the possible therapeutic use of EVs, authors should also mention about the use of EVs in cancer treatment
3)Liposomes have been also used in tissue engineering, this should be at least mentioned as a comparison with the use of EVs
Minor comment:
Reference number 17 has a mistake and should be updated
Author Response
We thank the Reviewer for the positive comment.
1) The aim of the section “Therapeutic potential of engineered EVs” (particularly from line 244) was to describe the strategies currently used for the EVs modification via several engineering techniques. In fact, the approaches that we have discussed in our work are aimed at manipulating the EVs in order to obtain a vehicle for chemical drugs or targeting molecules, for different therapeutic uses. However, we appreciate the Reviewer's suggestion and we have widened the discussion on this issue, reorganizing the section 3 titled “3. Therapeutic potential of EVs and exosomes: from cancer to regenerative medicine” (from 244).
2) We thank the Reviewer for the suggestion and we better discussed this aspect in from line 289.
3) We mentioned the use of liposomes in therapy (see lines 261-266).
All the minor points have been carefully addressed.

Reviewer 2 Report
Despite the article “On the choice for the extracellular vesicles for therapeutic use” is based on an important topic, this reviewer feels that this specific paper does not meet the main criteria for publication.
In fact, articles (original, review, mini-review, topical-review, commentary, opinion and so on) on EVs have increasingly saturated the scientific literature in the last 5 years. (Just as examples, see https://www.ncbi.nlm.nih.gov/pubmed/30471985 or https://www.ncbi.nlm.nih.gov/pubmed/30478831 or
https://www.mdpi.com/2077-0383/7/10/357 )
In this light, this article does not add no new information to what is already widely reported in literature.
Moreover, the narrative review should be avoided in reporting specific aims: authors should define an alternative, hopefully novel, topic involving this macro-concept of “the choice for the extracellular vesicles for therapeutic use”, trying to carry out with a systematic review the information they want to highlight in their work.
After an overall reading of this article, I can point out the following major points, to help authors for their future submission.
1. TITLE to be modified: “On the choice of the extracellular vesicles for therapeutic purposes”
2. TYPOS: article needs to be deeply revised for typos. Only for example, abstract starts as follows “Extracellular vesicles (EVs),” and the semicolon is not permitted between subject and verb.
3. Despite authors often describe to report “recent literature”, many papers are dated or not so related to the main topic. Moreover, authors should carefully consider not to cite their own entire bibliography, but of course this is only a friendly suggestion.
4. Some sentences (example - lines 148-151 - “All in all this paragraph has provided clear evidence that exosomes on one hand are a very important source of disease biomarkers but, on the other hand, they deliver molecules that may represent a danger when exosomes are thought to be exploited as a vehicle for therapeutic molecules, most of all exosomes derived from accessible sources such as plasma.”) are unclear, in a poor English, and do not improve the understanding of the final conclusive part of the paragraph.
5. Paragraph entitled “Therapeutic potential of engineered EVs” should be better structured. Authors should discuss their findings in the different branches of medicine. A table would be mandatory to make readers able to follow the numerous articles cited. It is not clear why authors use the term “engineered”, since it is not ever described TERM approach in the reported articles.
6. Figure 1 is not useful to this article: I suggest redrawing it with different aims or to remove it definitely.
7. A clear and useful conclusion section is not provided. Authors should give their expert opinion on what they have described. A clear section on the limits/cons of EVs, despite it is described in the abstract, it is not reported in the main text.
Author Response
Despite a large number of work regarding the therapeutic potential of EVs present in literature, the purpose of our work was to challenge the readers of IJMS on an uncovered issue such as the unmet enigma on which EVs to exploit in the therapy of the human diseases. We do not believe it is a so trivial issue inasmuch as there is not a clear agreement of the stuff worldwide. However, we appreciate the Reviewer's suggestions and we changed the text accordingly
1. We modified the title as suggest by Reviewer.
2., 3. and 4. We revised the text and improved the list of references in accordance with the Reviewer’s comments.
5. Starting from the line 259, we described the strategies for loading the EVs with non-native cargo. We state the existence of the two main approaches currently used, that are based on indirect modifications, via parental cells engineering; or on direct EVs manipulation. In the following discussion, regarding the examples reported in the literature, we also reported the modalities through which the EVs have been modified.
We thank the reviewer for the comment and we have also used the term "engineered" more appropriately.
6. We appreciate the Reviewer's suggestion and we revised the figure, outlining better the two different current approaches for the EVs manipulation in therapy, as discussed in the main text.
7. We changed the conclusion section in order to make clearer to the readers the pros and cons on the use of autologous and heterologous EVs in therapy. In addition, we discussed the future challenges that to be dealt with, in order to achieve a real development of new therapeutic approaches based on EVs.

Round 2
Reviewer 1 Report
The authors reviewed the manuscript which suits now for publication.
Author Response
Author reply: We thank the reviewer for his acceptance to publication.

Reviewer 2 Report
Despite Authors, in their rebuttal letter, appealed the reviewer’s first decision, claiming their article as a “perspective article” and not a “review”, however, in the body of their article, the authors reported “Considering the plethora of researches on personalized therapies has involved EVs as well, in particular those of nanosize, i.e. exosomes, through this review, we will predominantly focus the attention in exosomes, because they are currently the best characterized EVs .”
With the aim of re-evaluate this article for publication, this reviewer feels that the following points should be improved:
- Authors describe exosomes in the following way “Each class of EVs has specific markers, for example, tetraspanins such as Alix, Tsg101, CD9, CD63, CD81 and CD82, are typical exosomal markers, together with Heat Shock Proteins (Hsps) and MHC molecules.” However, in the light of their interest in using EVs for therapeutic purposes, they should discuss the role of CD146, as reported in the literature (Please, see and discuss: “Mesenchymal stem cell population isolated from the subepithelial layer of umbilical cord tissue. Patel AN, et al. Cell Transplant. 2013;22(3):513-9.”) and the potential role of CD146+ MSCs-derived EVs in future therapies (Please, see and discuss: “CD146 Expression Influences Periapical Cyst Mesenchymal Stem Cell Properties. Paduano F, et al. Stem Cell Rev. 2016 Oct;12(5):592-603”).
- In the light of future clinical applications, a paragraph should be added on the “regulatory issues” for RCTs on exosomes therapeutic use.
Author Response
Author reply:
- Point 1.: We thank the reviewer for these suggestions. In addressing the reviewer’s request we have added a sentence commenting the Paduano’s and Patel’s papers in the revised version (see line 70-71; 291-296).
- Point 2.: Actually, regarding the regulatory issues for RCTs on exosomes, there is not a clear institutional and regulatory framework on the clinical use of EVs, in both Europe and the US as well as in other Nations. This is the reason we are raising the problem.
Round 3
Reviewer 2 Report
Authors have addressed all the suggestions made by this reviewer. Article is currently more structured, with some interesting insights, and with figures/tables able to improve the overall readability.
I suggest acceptance.